

# Modeling the *cis*-regulatory modules of genes expressed in developmental stages of *Drosophila melanogaster*

Yosvany López[1,2], Alexis Vandenbon[3], Akinao Nose[4] and Kenta Nakai[1]

[1] Human Genome Center, The Institute of Medical Science, The University of Tokyo, Tokyo, Japan
[2] Department of Computational Biology, Graduate School of Frontier Sciences, The University of Tokyo, Chiba, Japan
[3] Immunology Frontier Research Center, Osaka University, Osaka, Japan
[4] Department of Complexity Science and Engineering, Graduate School of Frontier Sciences, The University of Tokyo, Chiba, Japan

## ABSTRACT

Because transcription is the first step in the regulation of gene expression, understanding how transcription factors bind to their DNA binding motifs has become absolutely necessary. It has been shown that the promoters of genes with similar expression profiles share common structural patterns. This paper presents an extensive study of the regulatory regions of genes expressed in 24 developmental stages of *Drosophila melanogaster*. It proposes the use of a combination of structural features, such as positioning of individual motifs relative to the transcription start site, orientation, pairwise distance between motifs, and presence of motifs anywhere in the promoter for predicting gene expression from structural features of promoter sequences. RNA-sequencing data was utilized to create and validate the 24 models. When genes with high-scoring promoters were compared to those identified by RNA-seq samples, 19 (79.2%) statistically significant models, a number that exceeds previous studies, were obtained. Each model yielded a set of highly informative features, which were used to search for genes with similar biological functions.

## INTRODUCTION

The control of the transcription mechanism is an important step in the regulation of gene expression (*Coulon et al., 2013*). Our understanding of how transcription factors act coordinately to regulate downstream genes is still insufficient. An abundance of studies have demonstrated the presence of common structural characteristics in the regulatory regions of genes expressed in the same physiological condition, cell or tissue (*López, Vandenbon & Nakai, 2014*; *Terai & Takagi, 2004*; *Vandenbon et al., 2008*; *Vandenbon & Nakai, 2010*; *Zhao, Schriefer & Stormo, 2007*).

Many approaches have been proposed to study *cis*-regulatory regions. A genome-wide prediction method was able to construct a *cis*-regulatory module and an epigenetic profile database, which collected transcription factor binding site information and nine types of

Corresponding author
Kenta Nakai,
knakai@ims.u-tokyo.ac.jp

epigenetic data (*Yang et al., 2014*). A graph-theoretic algorithm predicted *cis*-regulatory elements using chromatin immunoprecipitation (ChIP) datasets and effectively identified overrepresented combinatorial motif patterns (*Niu, Tabari & Su, 2014*). A computational approach was proposed for extracting and validating maps that relate regulatory sites to genes. These maps were able to detect long-range regulatory interactions (*O'Connor & Bailey, 2014*). Another study analyzed how *cis*-regulatory modules evolved in connection to natural selection and showed that complex modules use less specific binding motifs than smaller ones (*Stewart & Plotkin, 2013*). Although *cis*-regulatory regions typically contain multiple regulatory sites, the aforementioned studies were unable to predict common structural features within promoter regions. In addition, available ChIP-sequencing data is still limited to a few transcription factors under very specific conditions (*Mundade et al., 2014*; *Valouev et al., 2008*). There is a need for methodologies which can analyze common structural elements in promoters based on their sequence, and independently of other experimental data.

The developmental mechanism of *Drosophila melanogaster* constitutes an essential biological system, which allows us to better understand the transcription regulation of genes. It consists of three larval (first, second and third instar larva) and pupal stages before adult stage. The third instar larva molts into a pupa, which morphs into an adult fly. Adult structures are formed anew from two sets of cells: imaginal discs and histoblasts. Imaginal discs make the epidermal structures of the adult whereas histoblasts create the abdominal epidermis and internal organs of the adult (*Beira & Paro, 2016*; *Madhavan & Schneiderman, 1977*).

The regulatory regions of *Drosophila* have been widely analyzed. A previous study integrated genome-wide information related to transcription factor recruitment with *cis*-regulatory modules, histone modification and insulator binding from the whole *D. melanogaster* embryo during development. Consequently, enhancer occupancy and chromatin state were inferred to be able to predict spatio-temporal activity (*Wilczynski et al., 2012*). A high-resolution map has shown that different regulatory motifs influence the shape of transcription start site (TSS) distributions of those *D. melanogaster* promoters active in embryonic and adult stages (*Hoskins et al., 2011*). However, detailed insights about the regulatory mechanism of stage-specific genes are still insufficient.

Previously, we have modeled the promoters of several mouse and human tissues (*Vandenbon & Nakai, 2010*). We also analyzed the regulatory regions of antenna-expressed genes for common structural features, such as pairwise distance of motifs, orientation and distance of motifs relative to the TSS (*López, Vandenbon & Nakai, 2014*). This method was able to detect eight informative features, including characteristics related to positioning and orientation of motifs relative to the TSS, in the regions of antenna-expressed genes. Here, we have extended our former approach in order to shed light on the regulatory mechanism of genes expressed in 24 *D. melanogaster* developmental stages. The new models included additional features, such as presence of motifs without positional restriction and distance of motif pairs relative to the TSS. Contrary to a recent study, which has questioned whether or not the binding orientation of regulatory factors might be relevant (*Lis & Walther, 2016*), the features commonly present in our high-performing models included the presence of
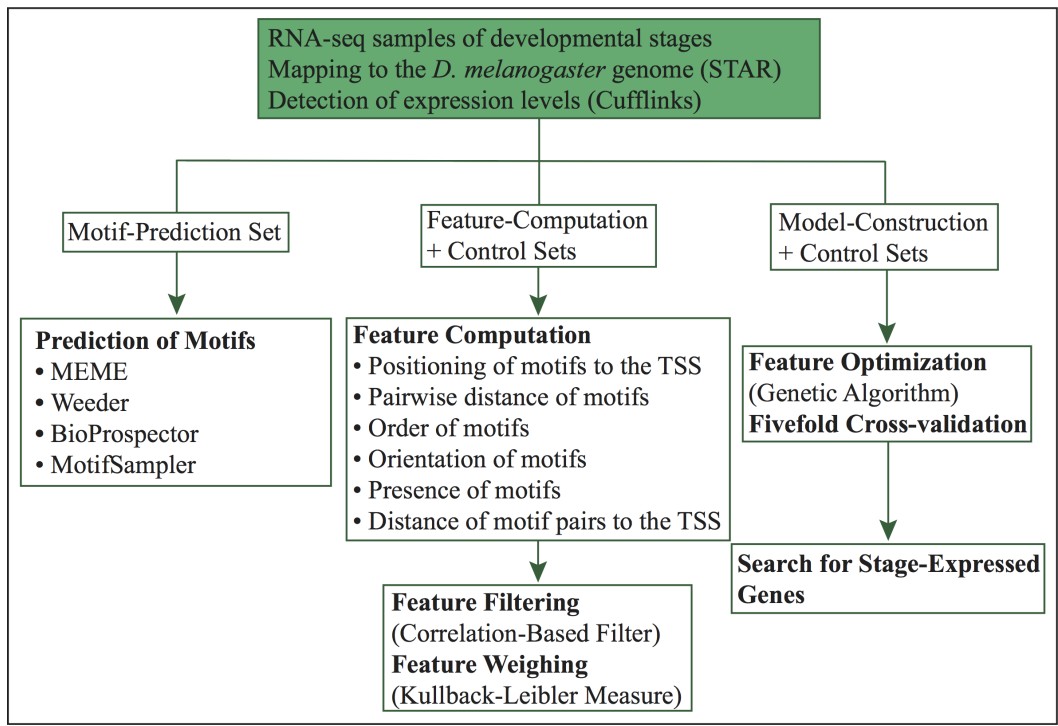

**Figure 1** **Workflow of our computational approach.**

motifs without positional restriction, the positioning of motifs relative to the TSS and the order of motifs in specific orientations. We further used the highly informative features of each model to search for co-expressed genes and considered RNA-seq data for validation purposes. When RNA-seq data was applied, we were able to successfully obtain 19 (79.2%) statistically significant models.

## MATERIALS AND METHODS

The current method (Fig. 1) is an extension of our previous approach (*López, Vandenbon & Nakai, 2014*). In this work, we predict specific promoter architectures for preferentially expressed genes in 24 different *D. melanogaster* developmental stages. For each stage, promoters more than 60% similar were removed and preferentially expressed genes were randomly distributed into three subsets (1) motif-prediction, (2) feature-computation, and (3) model-construction. The motif-prediction subset was used to predict *de novo* motifs. In the motif discovery step, we run four algorithms: MEME (*Bailey et al., 2006*), Weeder (*Pavesi, Mauri & Pesole, 2001*), BioProspector (*Liu, Brutlag & Liu, 2001*) and MotifSampler (*Thijs et al., 2002*). Redundant motifs were compared with the algorithm Tomtom (*Gupta et al., 2007*) and removed by retaining the motif with higher information content in each pair of matched motifs. The overrepresentation index (ORI) (*Bajic, Choudhary & Hock, 2003*) of non-redundant motifs was used to compute *p*-values and select the overrepresented motifs. The feature-computation subset was employed to compute structural features such as order, orientation, positioning of motifs relative to the TSS, pairwise positioning of

motifs, distance of motif pairs relative to the TSS and presence of motifs without positional restriction. Irrelevant characteristics were further removed with a correlation-based filter (*Yu & Liu, 2003*). The model-construction subset, used to reach highly informative features, was split into separate folds by fivefold cross-validation. A genetic algorithm, which maximizes fitness through crossovers and mutations, was trained using four folds and tested on the remaining one. Finally, we used the model of each stage to predict co-expressed genes with similar promoter structures. Each of these steps is introduced in more detail below.

## Gene expression datasets

Gene expression data (RNA-seq) that covered 25 developmental stages of *D. melanogaster* (Table S1) was obtained from the Model Organism Encyclopedia of DNA Elements (modENCODE) database (http://www.modencode.org). Each biological replicate sample was first visualized with the FastQC tool (*Andrews, 2010*) and quality thresholds were manually determined based on the distribution of mean sequence qualities. The FASTQ Quality Filter of FASTX-Toolkit (-v -p 80) (*Hannon-Laboratory, 2009*) was used to remove low-quality reads. The remaining reads were separately mapped to the *D. melanogaster* genome (r6.02) with the RNA-seq aligner STAR (*Dobin et al., 2013*). Cufflinks (supplied with reference annotation r6.02 from the FlyBase repository; parameter -G) (*Roberts et al., 2011*) was subsequently used to measure the expression level of each gene in fragments per kilobase of transcript per million mapped reads (FPKM). Expressed genes were identified as highly expressed in the corresponding stage relative to the other stages. In addition, the alignment of each replicate sample was independently used to assign sequence reads to genomic features (exons) with the featureCounts program, which is part of the Rsubread package (*Liao, Smyth & Shi, 2014*). The read counts were then input to the edgeR package (*Robinson, McCarthy & Smyth, 2010*) for determining differentially expressed genes per stage. For each developmental stage, we selected an initial set of expressed genes whose expression level was >1 FPKM, and were differentially expressed at a false discovery rate of 5%. It is worth noting that there might be overlapping of expressed genes among developmental stages. On the other hand, genes with expression level of 0 or adjusted *p*-value of 1 were regarded as non-expressed genes and grouped into a control set. Because small gene sets could heavily bias downstream analyses in each model, we decided to retain all sets with at least 55 genes for further analysis. As a result, 24 developmental stages were ultimately considered (Table S2).

The *D. melanogaster* genome (r6.02) was downloaded from the FlyBase repository (*Marygold et al., 2013*) and the genomic stretch from 1.5 kbp upstream to 500 bp downstream of the TSS was retrieved as promoter region (*Roy et al., 2010*). Multiple promoters with similar sequences tend to bias the motif prediction and feature extraction steps. To avoid this, we removed the expressed genes whose regions were >60% similar with the program cd-hit (clustering threshold = 0.6; word length = 3) (*Fu et al., 2012*). For each developmental stage, we then defined a set of preferentially expressed genes, and their promoter regions. Below, we will use these promoters to predict common regulatory motifs.

## Final gene sets

Each set of stage-expressed genes was split into three independent sets: (1) motif-prediction, (2) feature-computation, and (3) model-construction. To do this, we randomly distributed each initial gene set so that 40% assigned to motif-prediction, 20% to feature-computation, and 40% to model-construction. The motif-prediction set was used for predicting *de novo* motifs, the feature-computation set was employed to compute structural features, and the model-construction set was used for selecting highly informative features in the promoters of expressed genes. Moreover, the aforementioned control set was evenly split into two separate groups. One subset was employed as control for computing structural features, whereas the other was utilized for constructing the models. To assess the performance of our models we used a fivefold cross-validation scheme, whereby the model-construction set was evenly split into five folds for running the genetic algorithm (explained below).

## Prediction, selection and comparison of motifs

The motif discovery algorithms MEME (*Bailey et al., 2006*), Weeder (*Pavesi, Mauri & Pesole, 2001*), BioProspector (*Liu, Brutlag & Liu, 2001*) and MotifSampler (*Thijs et al., 2002*) were employed for predicting *de novo* motifs in the motif-prediction set of promoters. We first masked the promoters of each motif-prediction set with the RepeatMasker program (*Smit, Hubley & Green, 2015*) and separately predicted motifs for every developmental stage. We used MEME (*Bailey et al., 2006*) to predict 6- to 12-bp motifs with any number of sites per sequence on both strands, and Weeder (*Pavesi, Mauri & Pesole, 2001*) to search for 6-bp motifs with one mutation, 8-bp motifs with two and three mutations, 10-bp motifs with three and four mutations, and 12-bp motifs with four mutations on both strands. For BioProspector (*Liu, Brutlag & Liu, 2001*) and MotifSampler (*Thijs et al., 2002*), the top five motifs with lengths of 6–12 bp were searched on both strands, and predictions with BioProspector (*Liu, Brutlag & Liu, 2001*) were performed 100 times.

Each collection of *de novo* motifs was merged with known motifs in TRANSFAC (Insecta) (190 matrices) (*Matys et al., 2006*) and JASPAR Core Insecta (131 matrices) (*Mathelier et al., 2014*) databases. The motif comparison algorithm Tomtom (minimal overlapping = 1; distance function = euclidean) (*Gupta et al., 2007*) was employed for removing redundant motifs. Per pair of similar motifs (*p*-value $\leq$ 0.001), the motif with higher information content (*Stormo & Fields, 1998*) was retained. Also, motifs that did not show any similarity were regarded for further analysis.

For remaining motifs, in addition to calculating the ORI (*Bajic, Choudhary & Hock, 2003*), one million random ORI values were computed. To do so, all the *D. melanogaster* promoter regions (1.5 kbp upstream and 500 bp downstream of the TSS) and non-promoter regions (from 2 kbp to 4 kbp downstream of the TSS) were scanned for motifs. Subsequently, one million sets of the same size as the motif-prediction set were randomly selected for computing a reference ORI distribution. The ratio of random values greater than the original ORI value was calculated as *p*-value. Finally, motifs with *p*-values <0.01 were considered to be overrepresented, and used for the generation of features.

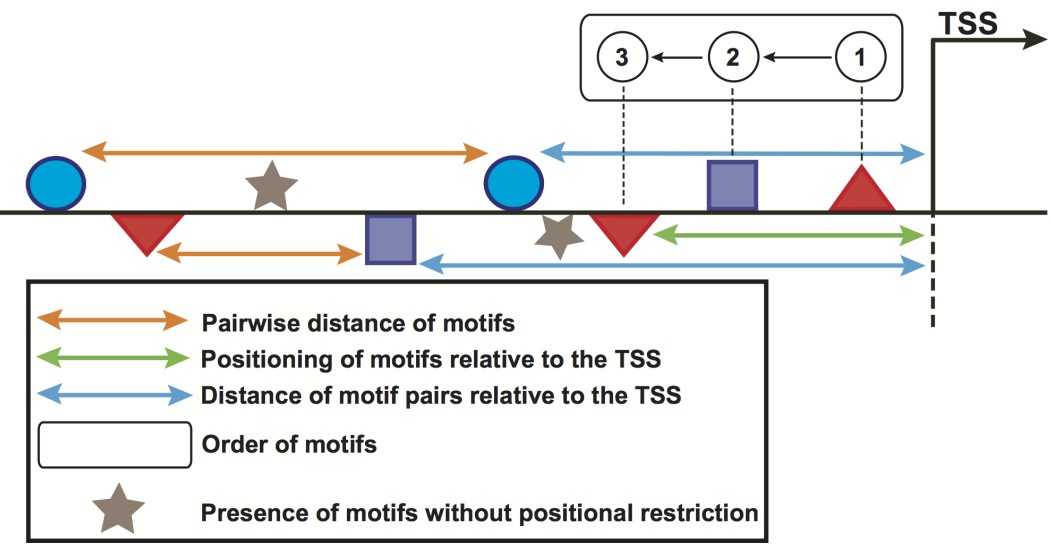

**Figure 2 Schematic representation of the computed features in promoter regions of stage-expressed genes.** Geometrical forms above/below the horizontal line represent the orientation of sequence motifs towards plus/minus strands.

## Computation and filtering of features

We used the feature-computation and the corresponding control sets to compute a feature collection from the motifs predicted in the previous step. Six structural features, namely orientation, positioning of motifs relative to the TSS, pairwise distance of motifs, presence of motifs without positional restriction and distance of motif pairs relative to the TSS (Fig. 2) are considered here. To compute these features, the regions of genes in the above-mentioned sets were scanned with 100-bp windows in steps of 100 bp. In doing so, the order of motifs was calculated independently of strand orientation, for regions both upstream and downstream of the TSS. For the positioning of motifs relative to the TSS, we placed the window at the TSS and scanned both upstream and downstream regions. For the pairwise distance of motifs, one of the motifs was regarded as the starting point whereas the presence feature considered any occurrence of the motif without positional restrictions. For the distance of motif pairs relative to the TSS, we regarded the distance of the closest motif to the TSS. These features computed in the promoters of genes in the feature-computation and control sets were further binarized. Each region was represented as one vector indicating presence (1) or absence (0) of the features. A binary matrix was thus built for each stage. Because the feature set could be relatively large and contain a great deal of redundancy, we utilized the above matrix to preprocess the feature collection with a correlation-based filter (*Yu & Liu, 2003*). This filter uses a measure known as "symmetrical uncertainty", which removes features with low correlation by using the promoter regions in the feature-computation (positive class, 1) and the control (negative class, 0) sets. More information on this section can be found in *López, Vandenbon & Nakai (2014)*.

## Optimization of features

The previous step generates features which are likely to contain a certain degree of redundancy. To remove redundant characteristics, the feature collection of each model was further reduced to a smaller subset of highly informative features. To do this, we made use of two different scores and a genetic algorithm. One score (Fscore) takes into account the presence/absence of features and was introduced to assess the fitness of solutions in the genetic algorithm. For each model, the genetic algorithm was run with the model-construction and control sets as input. In the genetic algorithm, a population equal to the number of training promoters in the model-construction and control sets was regarded. In each individual, a "chromosome" was used to encode the subset of selected features. The algorithm utilized a uniform crossover and a mutation probability of 0.05. The Fscore, which considers both precision and recall, was defined as follows,

$$Fscore = 2 * \frac{precision * recall}{precision + recall} \tag{1}$$

$$precision = \frac{TP}{TP + FP} \tag{2}$$

$$recall = \frac{TP}{TP + FN} \tag{3}$$

where $TP$, $FP$ and $FN$ stand for true positives, false positives and false negatives. True positives and false negatives are the regulatory regions in the model-construction set, which contain at least one feature, or no features, respectively. False positives are the promoter regions, with at least one feature, in the corresponding control set. As stopping criteria, we used the following conditions: the genetic algorithm was iterated at most 10,000 times, or stopped when a solution with $Fscore \geq 0.8$ was reached. In addition, we added an extra stop criterion based on a second scoring function. Namely, if >50% of the positive regions (model-construction set) were scored higher than 90% of the negative regions (control set). For this, we scored the promoter regions using the following equation,

$$score_{region} = \sum_i w_i * n_i \tag{4}$$

where $score_{region}$ takes into consideration the number of times the informative features appear in the region. In other words, the number of occurrences ($n_i$) and the weight ($w_i$) of each feature $i$ in the feature set were multiplied and summed up. The weight $w_i$ was computed as follows,

$$w_i = \frac{\sum_{j|i} P(o_{ij}) * D_{KL}(C|o_{ij})}{-\sum_{j|i} P(o_{ij}) * \log(P(o_{ij}))} \tag{5}$$

where $o_{ij}$ represents the promoter region with the $j$ value (presence/absence) of feature $i$, $P(o_{ij})$ is the frequency of promoter regions $o_{ij}$ and $D_{KL}(C|o_{ij})$ is the Kullback–Leibler measure of class $C$ (positive and negative classes) given the promoter regions $o_{ij}$. The positive and negative classes comprise the promoters of genes in the feature-computation and control sets, respectively. The Kullback–Leibler measure (*Lee, Gutierrez & Dou, 2011*),

which assigns greater values to features often present in the promoter regions of stage-expressed genes, was calculated by

$$D_{KL}\left(C|o_{ij}\right) = \sum_{c} P\left(c|o_{ij}\right) \log\left(\frac{P\left(c|o_{ij}\right)}{P\left(c\right)}\right) \qquad (6)$$

where $P(c)$ is the probability of class $c$ and $P(c|o_{ij})$ represents the probability of class $c$ given the promoter regions $o_{ij}$. To avoid meaningful features being assigned very low weights, we omitted the normalization step of summing to unity as done in our previous study (*López, Vandenbon & Nakai, 2014*).

Both the model-construction and the corresponding control sets were split into separate folds by fivefold cross-validation. The genetic algorithm was trained using four folds and tested on the remaining one. This process was repeated five times and the overall Fscore of the model was the average of the highest-performing solution in each test fold. For each model, we then regarded those characteristics present in at least four of the five highest-scoring individuals as its highly informative features.

## Validations

Validations were separately conducted for assessing the accuracy of each model in predicting stage-expressed genes with similar promoter architectures. Each collection of informative features was used to score (using Eq. (4)) the entire promoter set of *D. melanogaster*. We excluded the promoters of genes (both stage-expressed and control ones) used for training the models, and retrieved the top 100 genes with high-scoring promoter regions per developmental stage. In addition, 24 independent RNA-seq samples (Table S3) that covered the developmental stages were downloaded from the modENCODE database (http://www.modencode.org). Each biological replicate was separately mapped to the *D. melanogaster* genome (as explained in the section "Gene Expression Datasets"), and genes with FPKM >1 were considered to be expressed. The number of overlapping genes (i.e., how many high-scoring genes are also detected in the corresponding RNA-seq sample) was checked. Thus, for each developmental stage we obtained an independent test set to evaluate the accuracy of our promoter models. Furthermore, for each stage we corroborated whether overrepresented motifs associated with stage-expressed genes were located inside experimentally characterized *cis*-regulatory elements in REDfly database (*Gallo et al., 2011*). For this, we downloaded the entire collection of "Reporter Constructs" and "TFBS" from REDfly repository (*Gallo et al., 2011*), and checked the genomic coordinates of predicted motifs and annotated elements. In order to perform a quantitative assessment, we also downloaded ChIP-seq data for two well-known transcription factors: dorsal (Gene Expression Omnibus ID SRR2031908) and snail (Gene Expression Omnibus ID SRR2031905), which play important roles during dorsal-ventral axis formation in the *Drosophila* embryo (*Zeitlinger et al., 2007*). This data was separately mapped to the *Drosophila* genome with Bowtie2 (*Langmead & Salzberg, 2012*) and regarding two mismatches. Duplicates were removed and potential binding sites were identified with the peak caller MACS2 (FDR cut-off = 0.01) (*Yong et al., 2008*). For each transcription factor, we then overlapped its detected binding sites with sequence motifs predicted by the models of stages 0–2 h and 2–4 h. This

step was carried out with the bedmap program (*Neph et al., 2012*), and strictly regarding an overlapping when the entire sequence motif lay inside detected binding sites. Moreover, Fscores were computed with one million sets in which labels were randomly shuffled. The ratio of random Fscores greater than the original Fscore was calculated as a *p*-value. All the resulting uncorrected *p*-values were further adjusted for multiple test comparisons with the R function "p.adjust" (method = "fdr"). Consequently, models showing corrected *p*-values < 0.01 were considered to be statistically significant.

## RESULTS AND DISCUSSION

In order to understand the promoter architecture of co-expressed genes, we analyzed structural features in promoter sequences of genes expressed in *D. melanogaster* cell types across different developmental stages. This work proposes an extended approach, which combines a correlation-based filter and a genetic algorithm, for detecting highly informative sets of common structural features.

### Predicted stage-related motifs

We used a part of the training promoters (motif-prediction set) for *de novo* prediction of enriched motifs in the region surrounding the TSS of stage-expressed genes. The predicted motifs and the known ones in TRANSFAC (*Matys et al., 2006*) and JASPAR (*Mathelier et al., 2014*) databases were processed for removing redundant motifs, and obtaining the overrepresented ones per stage (Table S4).

The matrices of enriched motifs were compared to those of known motifs in the two aforementioned databases (*Mathelier et al., 2014*; *Matys et al., 2006*) with the algorithm Tomtom (*Gupta et al., 2007*). We found some enriched motifs, which resemble the binding preferences of known factors of importance in *D. melanogaster* development (Table S5). One example is a motif found in stages embryo 0–2 h and embryo 2–4 h, which resembles the binding motif of regulator dl (dorsal). dl plays a key role in establishing dorsal-ventral polarity at early stages of *Drosophila* development. Initially located in the cytoplasm, it is transported to the nucleus approximately 90 min after fertilization where it functions as a morphogen (*Rushlow et al., 1989*; *Steward, 1989*). Another instance is the detection of a motif in stage embryo 0–2 h, which shows similarity to that recognized by transcription factor sna (snail). This fits well with reports that the regulatory function of sna is initiated around two hours after egg laying (*Campos-Ortega & Hartenstein, 1997*). A motif bearing resemblance to regulator Cf2 (Chorion factor 2) was discovered in stage embryo 12–14 h. Cf2 has been shown to regulate cell fate determination during oogenesis, and appears when skeletal myoblast fusion is taking place (*Bagni et al., 2002*). Although transcripts encoding the regulator Cf2 are subject to alternative splicing, and a single gene could encode sets of genes in different developmental stages (*Hsu et al., 1992*), it was not found in other stages. Furthermore, motifs recognized by the regulator croc (crocodile) were detected in three stages: embryo 12–14 h, L3 stage larvae, and adult female eclosion +30 days. This regulatory factor is expressed in the head anlagen of the blastoderm embryo. Besides it also establishes a specific head skeletal structure from intercalary segments at later stages of embryogenesis (*Häcker et al., 1995*). Additionally, we plotted the expression of transcription factor genes

per stage in an attempt to check whether factors linked to known motifs could be regulating stage-expressed genes. For each transcription factor, Fig. 3 illustrates the expression of its gene in the stage it was detected. Expression levels in the remaining stages are not indicated so that these cells are shown in grey. As Fig. 3 depicts, 10 out of 14 genes encoding regulatory factors are highly expressed in the corresponding stage. Among the genes showing low expression are: *croc* (crocodile) only in stages L3 stage larvae and adult female eclosion +30 days, *lbe* (ladybird early) in stages embryo 22–24 h, white prepupae, white prepupae +24 h and adult female eclosion +30 days, *Pph13* (PvuII-PstI homology 13) in stage embryo 20–22 h, and *ftz* (fushi tarazu) in stage adult female eclosion +30 days. These findings support the ability of our motif prediction tools to correctly detect stage-related factors, which might actively regulate genes expressed in *Drosophila* developmental stages.

## Performance of the models

The enriched motifs were used to create a collection of features per stage. These characteristics were first filtered and then codified into a genetic algorithm for selecting highly informative features. The resulting Fscores of the models ranged from 0.596 to 0.992 (Table 1). The models of embryo 12–14 h, white prepupae +24 h and pupae achieved the highest average (fivefold cross-validation) performances with 0.905, 0.912 and 0.992, respectively. For each stage, the best individuals (sets of features) of the five cross-validation runs were retrieved, and features present in at least four individuals were regarded to be highly informative features. Figure 4 depicts the informative features of the three models with the highest performance. As it shows, features related to the positioning of motifs relative to the TSS were detected by two models (Figs. 4B–4C), whereas the three models yielded the order of motifs as a highly informative feature. As a result, 20 (83.3%) models turned out to be statistically significant (adjusted $p$-value $< 0.01$) (Table 1; Fig. 5B). In order to avoid any confusion, it is worth noting that this number of significant models tended to slightly decrease when validated with RNA-seq data (refer to section 'Validation of the Models').

The models comprised an average of 6.04 motifs, with a range of 2 (white prepupae) to 10 (embryo 0–2 h, embryo 22–24 h and adult female eclosion +1 day) motifs (Table S4).

Furthermore, we created several models by gradually including different kinds of features. We first built simple models only regarding the presence of motifs, and subsequently more complex models comprising features such as orientation, positioning relative to the TSS, pairwise distance and order of motifs (Figs. 5A and 5B; Table S6). Although the models that comprised features such as presence (with or without orientation) of motifs performed very poorly, there was a clear tendency for performance to increase when models were allowed to contain more complex features (M-1 to M-4 in Fig. 5B). However, the addition of motif order (M-5 in Fig. 5B) did not improve performance any further. Finally, we trained the models by leaving out the orientation of motifs (M-7 in Fig. 5B), and though the overall performance (62.5%) was reasonable it was not greater than that of M-4 and M-6 models.

In general, the models seem to best perform when features like presence, orientation, positioning relative to the TSS and pairwise positioning of motifs, or all the six features, were included. In both cases, we were able to obtain 20 (83.3%) significant models. However,

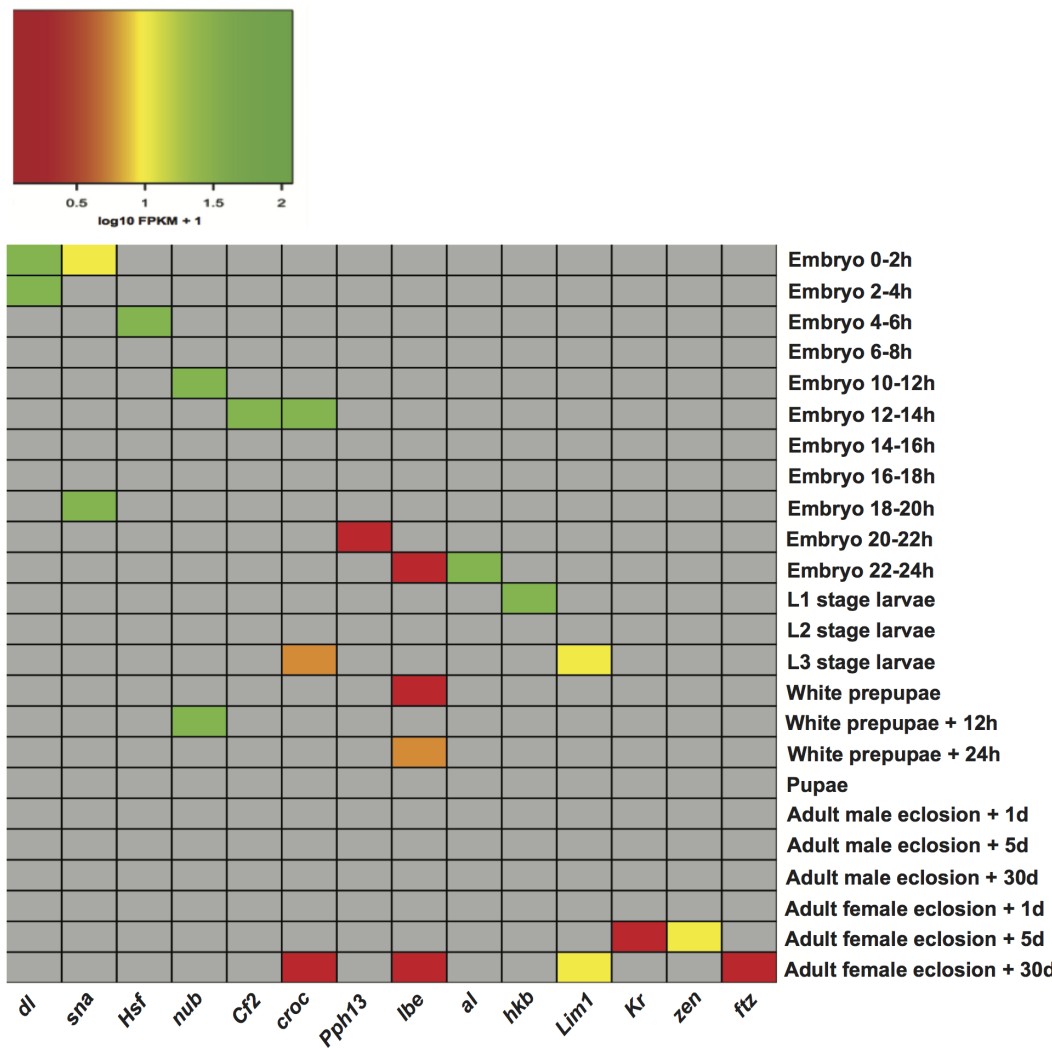

**Figure 3  Heatmap of the expression level of transcription factor genes in specific developmental stages.** *dl* (dorsal), *sna* (snail), *Hsf* (Heat shock factor), *nub* (nubbin), *Cf2* (Chorion factor 2), *croc* (crocodile), *Pph13* (PvuII-PstI homology 13), *lbe* (ladybird early), *al* (aristaless), *hkb* (huckebein), *Lim1* (LIM homeobox 1), *Kr* (Kruppel), *zen* (zerknullt), *ftz* (fushi tarazu).

when the order of motifs was considered, the overall performance slightly decreased. Similarly, the presence and orientation of motifs sharply decreased the overall performance of the models (Figs. 5A and 5B; Table S6).

Subsequently, we examined whether highly expressed genes in different developmental stages could be under the control of similar regulatory elements. We therefore used the model of each developmental stage to score genes expressed in the other stages (Fig. 6), and evaluated the significance of the scores (Student's $t$-test). Although the models of white prepupae, pupae and adult female eclosion $+30$ days consistently assigned high scores to genes expressed in many stages, apparently showing a lack of specificity or mostly identifying housekeeping genes, other models such as embryo 10–12 h and L1 stage larvae tended to highly score genes in nearby stages (Fig. 6).

**Table 1 Performance of the computational models regarding all the six features.** For each model, developmental stage and measures such as Fscore, accuracy (Acc), sensitivity (Sn), specificity (Sp), *P*-Value and False Discovery Rate (FDR) are shown.

| Developmental stage | *F*score | Acc | Sn | Sp | *P*-value | FDR |
|---|---|---|---|---|---|---|
| Embryo 0–2 h | 0.681 | 0.614 | 0.829 | 0.400 | 0.00E+00 | 0.00E+00[*] |
| Embryo 2–4 h | 0.707 | 0.640 | 0.847 | 0.433 | 3.70E−03 | 5.22E−03[*] |
| Embryo 4–6 h | 0.695 | 0.642 | 0.808 | 0.475 | 6.43E−03 | 7.88E−03[*] |
| Embryo 6–8 h | 0.715 | 0.680 | 0.800 | 0.560 | 6.57E−03 | 7.88E−03[*] |
| Embryo 10–12 h | 0.655 | 0.610 | 0.762 | 0.457 | 0.00E+00 | 0.00E+00[*] |
| Embryo 12–14 h | 0.905 | 0.894 | 1.000 | 0.789 | 0.00E+00 | 0.00E+00[*] |
| Embryo 14–16 h | 0.599 | 0.524 | 0.714 | 0.333 | 2.94E−01 | 2.94E−01 |
| Embryo 16–18 h | 0.723 | 0.675 | 0.850 | 0.500 | 1.27E−03 | 1.91E−03[*] |
| Embryo 18–20 h | 0.596 | 0.544 | 0.667 | 0.422 | 2.10E−01 | 2.19E−01 |
| Embryo 20–22 h | 0.659 | 0.593 | 0.781 | 0.405 | 2.91E−02 | 3.33E−02 |
| Embryo 22–24 h | 0.669 | 0.562 | 0.889 | 0.235 | 0.00E+00 | 0.00E+00[*] |
| L1 stage larvae | 0.605 | 0.600 | 0.611 | 0.589 | 0.00E+00 | 0.00E+00[*] |
| L2 stage larvae | 0.696 | 0.647 | 0.808 | 0.485 | 7.95E−04 | 1.27E−03[*] |
| L3 stage larvae | 0.868 | 0.844 | 1.000 | 0.688 | 0.00E+00 | 0.00E+00[*] |
| White prepupae | 0.819 | 0.761 | 1.000 | 0.522 | 0.00E+00 | 0.00E+00[*] |
| White prepupae +12 h | 0.780 | 0.791 | 0.739 | 0.844 | 0.00E+00 | 0.00E+00[*] |
| White prepupae +24 h | 0.912 | 0.921 | 0.843 | 1.000 | 0.00E+00 | 0.00E+00[*] |
| Pupae | 0.992 | 0.992 | 0.985 | 1.000 | 0.00E+00 | 0.00E+00[*] |
| Adult male eclosion +1 day | 0.637 | 0.556 | 0.778 | 0.333 | 4.41E−02 | 4.81E−02 |
| Adult male eclosion +5 days | 0.763 | 0.748 | 0.810 | 0.686 | 0.00E+00 | 0.00E+00[*] |
| Adult male eclosion +30 days | 0.681 | 0.552 | 0.956 | 0.148 | 0.00E+00 | 0.00E+00[*] |
| Adult female eclosion +1 day | 0.692 | 0.646 | 0.795 | 0.498 | 5.38E−04 | 9.22E−04[*] |
| Adult female eclosion +5 days | 0.694 | 0.613 | 0.880 | 0.347 | 5.02E−03 | 6.69E−03[*] |
| Adult female eclosion +30 days | 0.739 | 0.642 | 1.000 | 0.284 | 0.00E+00 | 0.00E+00[*] |

**Notes.**
*Statistically significant models are indicated by superscript asterisks.

## Frequency of features

To assess the importance of features in describing each promoter set, we also analyzed the frequency of features in the created models. To do so, we manually grouped all the informative features into the six types of features regarded in this study (refer to 'Materials and Methods'). These six feature collections are: presence of motifs regardless of orientation (Presence), presence and orientation of motifs (Presence & Orientation), position of motifs relative to the TSS, pairwise distance of motifs, order and distance of two motifs relative to the TSS (Fig. 5C). For each feature, we checked its presence in promoters of the model-construction set of the stage it was detected in. Consequently, the features that describe presence and pairwise positioning of motifs towards a specific orientation were most frequent, representing 28% and 25% of all the features. Both features were followed by those related to positioning of motifs relative to the TSS (21% of the total) and distance of two motifs to the TSS (15% of the total). On the other hand, order and orientation of motifs accounted for only 11% and 0% of all the features, respectively (Fig. 5C).

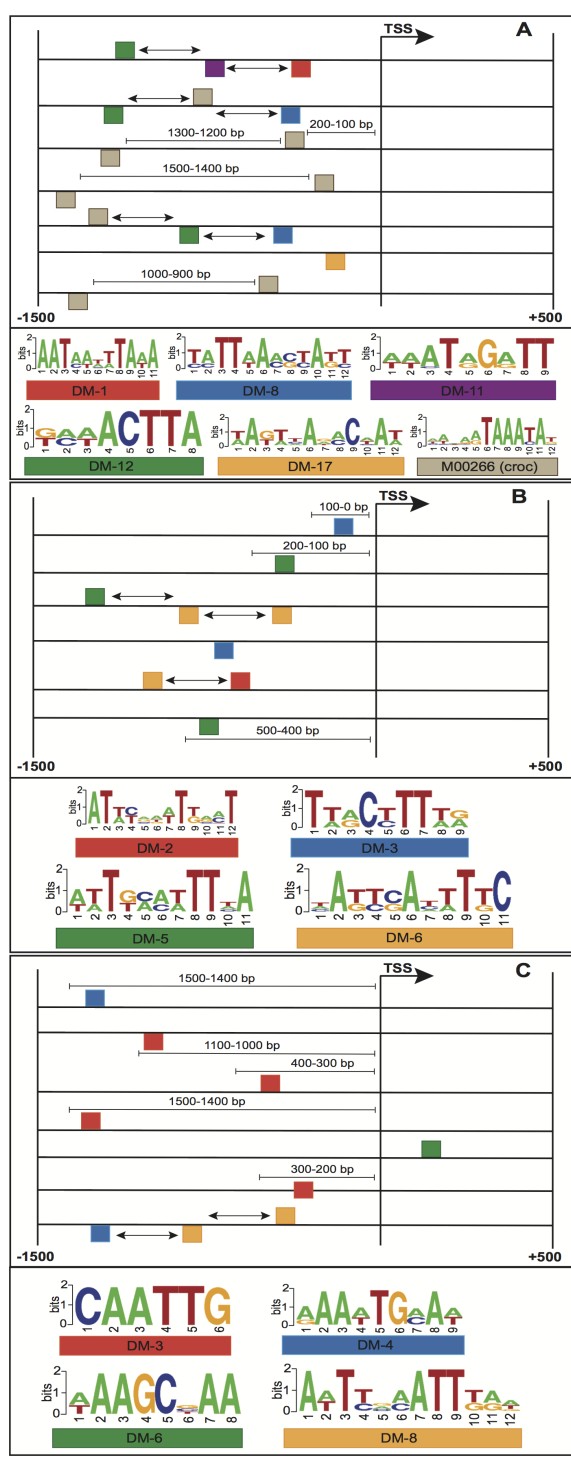

**Figure 4  Structural features of three models with the highest performance, (A) embryo 12–14 h, (B) white prepupae +24 h and (C) pupae.** Squares above/below the horizontal line indicate the DNA strand where the motif is located. Arrows represent features related to order of motifs.

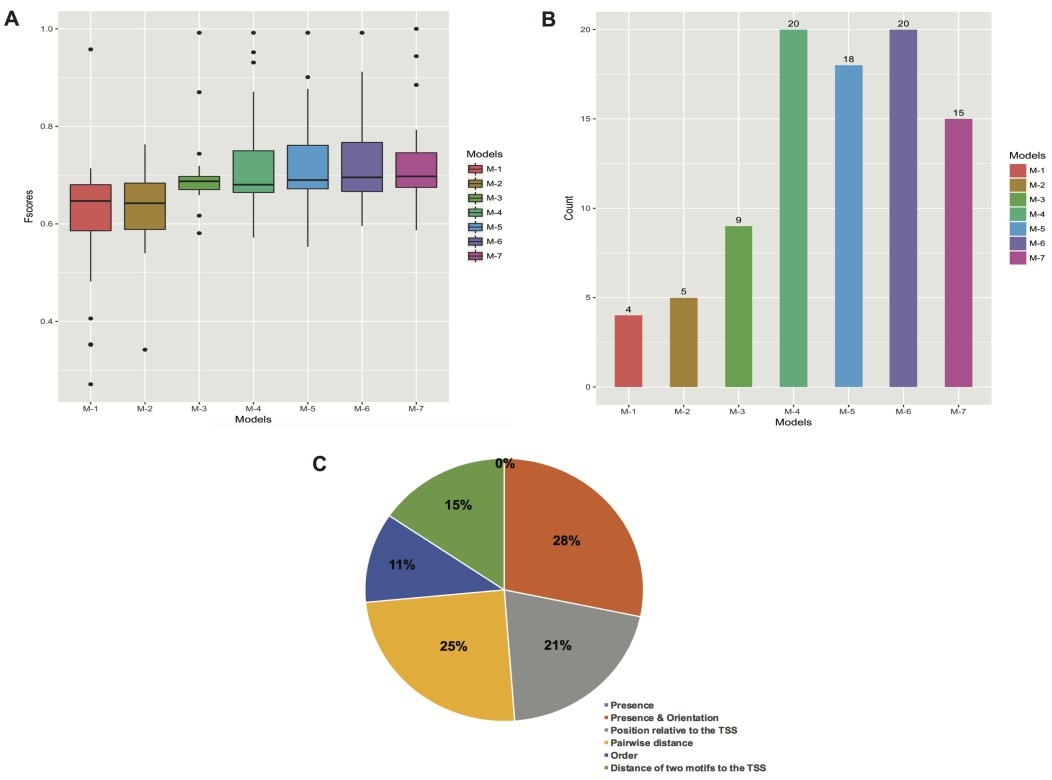

**Figure 5** **Performance of the models with different types of features. (A) Boxplots of Fscores. (B) Bar plots of the number of significant models. (C) Frequency of informative features in all the models.** The M's stand for simple to more complex models: M-1 (presence of motifs), M-2 (presence and orientation of motifs), M-3 (presence, orientation and positioning of motifs relative to the TSS), M-4 (presence, orientation, positioning of motifs relative to the TSS and pairwise distance of motifs), M-5 (presence, orientation, positioning of motifs relative to the TSS, pairwise distance and order of motifs), M-6 (all the features), and M-7 (presence, positioning of motifs relative to the TSS and pairwise distance of motifs). The "0%" means that no features related to presence of motifs regardless of orientation were obtained.

## Validation of the models

Each set of informative features was subsequently used to score the whole promoter set of *D. melanogaster*, and to search for stage-expressed genes with similar promoter structures. We excluded the promoters of genes (both stage-expressed and control ones) used to construct the models, and selected the top 100 genes with high-scoring regions per developmental stage. Independent RNA-seq samples (Table S3) were mapped to the *D. melanogaster* genome and expressed genes were retrieved. For each developmental stage, the number of overlapping genes between high-scoring ones and genes confirmed by the corresponding RNA-seq sample was analyzed. As Table 2 illustrates, for 17 models, $\geq 70$ out of the top 100 high-scoring genes were stage-expressed. The threshold for choosing between expressed and non-expressed genes was set to FPKM $>1$. This cut-off was also changed to FPKM $>2$ and FPKM $>3$ in order to verify how different FPKMs would affect the results. Although there was a decrease in the number of stage-expressed genes, $>50\%$ of the models still showed statistical significance for FPKM $>2$ (Table S7). We further downloaded experimentally validated *cis*-regulatory elements from the REDfly database

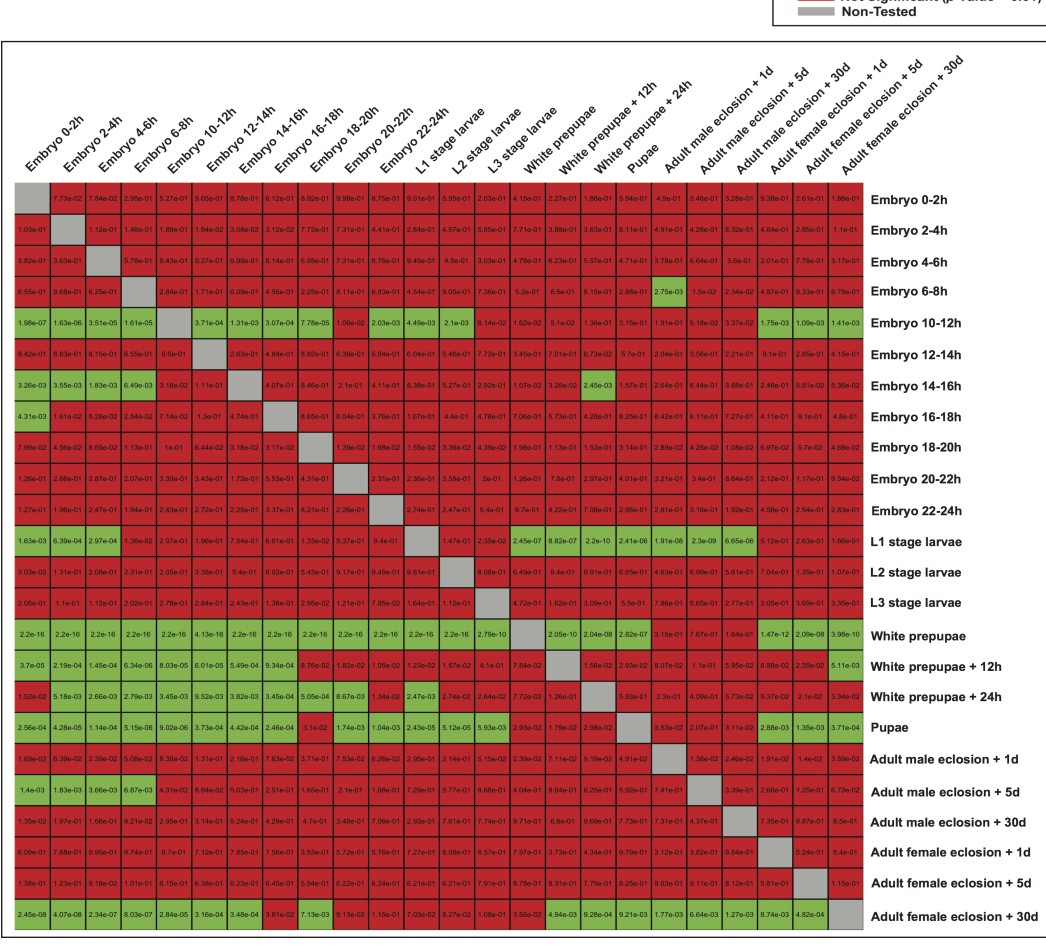

**Figure 6** **Heatmap of *p*-values indicating the ability of one model to characterize the promoter regions of genes expressed in the other stages.** For each stage, all the genes were scored by the model of another stage (the rows indicate the models and the columns are the score sets). The scoring of expressed and non-expressed genes was evaluated with the Student's *t*-test.

(*Gallo et al., 2011*), and checked whether predicted motifs associated with stage-expressed genes were located inside validated regions. This analysis resulted in many of our predicted sites lying within experimentally validated elements (Data S1). One of the best-understood regulatory processes in the *Drosophila* embryo is the dorsal-ventral axis formation. It occurs early in embryogenesis, and the role of enhancers in controlling patterning and cell fate specification has been previously studied (*Zeitlinger et al., 2007*). Therefore, we further downloaded ChIP-seq data of two transcription factors: dorsal and snail, in order to conduct a more quantitative evaluation. For each transcription factor, we then overlapped detected binding sites with sequence motifs predicted by the models of stages embryo 0–2 h and 2–4 h. As a result, we were able to predict many sequence motifs located inside detected sites. The factor snail contributes to the establishment of the mesoderm primordium in the early *Drosophila* embryo in addition to regulating morphogenesis and differentiation of the mesoderm. Our analysis detected snail occupancies inside the promoter regions of

**Table 2  Number of genes among the top 100 high-scoring genes, which were also confirmed by RNA-seq data.** Genes used for designing the models were not scored. The count of expressed and high-scoring genes, the genome-wide fraction of expressed genes (FPKM > 1) and *p*-value from the hypergeometric test are included.

| Developmental stage | Count | Expression fraction | *P*-value |
|---|---|---|---|
| Embryo 0–2 h | 56 | 0.397 | 7.13E−04[*] |
| Embryo 2–4 h | 45 | 0.437 | 4.33E−01 |
| Embryo 4–6 h | 61 | 0.468 | 2.92E−03[*] |
| Embryo 6–8 h | 50 | 0.462 | 2.54E−01 |
| Embryo 10–12 h | 64 | 0.516 | 7.99E−03[*] |
| Embryo 12–14 h | 70 | 0.527 | 3.15E−04[*] |
| Embryo 14–16 h | 71 | 0.552 | 8.37E−04[*] |
| Embryo 16–18 h | 76 | 0.592 | 3.07E−04[*] |
| Embryo 18–20 h | 62 | 0.665 | 8.54E−01 |
| Embryo 20–22 h | 75 | 0.597 | 9.69E−04[*] |
| Embryo 22–24 h | 79 | 0.597 | 3.18E−05[*] |
| L1 stage larvae | 75 | 0.587 | 4.78E−04[*] |
| L2 stage larvae | 73 | 0.599 | 4.13E−03[*] |
| L3 stage larvae | 89 | 0.691 | 2.32E−06[*] |
| White prepupae | 77 | 0.646 | 5.15E−03[*] |
| White prepupae +12 h | 74 | 0.678 | 1.09E−01 |
| White prepupae +24 h | 78 | 0.626 | 7.21E−04[*] |
| Pupae | 85 | 0.692 | 2.13E−04[*] |
| Adult male eclosion +1 day | 84 | 0.702 | 1.09E−03[*] |
| Adult male eclosion +5 days | 85 | 0.708 | 6.88E−04[*] |
| Adult male eclosion +30 days | 91 | 0.722 | 3.33E−06[*] |
| Adult female eclosion +1 day | 73 | 0.599 | 4.21E−03[*] |
| Adult female eclosion +5 days | 57 | 0.542 | 3.20E−01 |
| Adult female eclosion +30 days | 71 | 0.533 | 2.15E−04[*] |

**Notes.**
[*]Statistically significant models are indicated by superscript asterisks.

well-known genes in dorso-ventral pattern formation. One of these target genes was *stumps* (FlyBase ID FBgn0020299) (*Casal & Leptin, 1996*). Another gene was *Socs36E* (FlyBase ID FBgn0041184), which follows an expression pattern similar to that of the *Drosophila* JAK/STAT pathway ligand unpaired. *Socs36E* is upregulated when the ectopic activation of JAK/STAT pathway occurs during embryonic and imaginal development (*Karsten, Häder & Zeidler, 2002*). Snail binding sites were also found in the regulatory region of *tailup* (FlyBase ID FBgn0003896), whose early function in the development of imaginal wing disc has been studied (*Navascués & Modolell, 2007*). The factor dorsal also plays key roles in the regulation of specific genes. For instance, we identified dorsal occurrences in the promoters of *Cyp310a1* (FlyBase ID FBgn0032693) and *wntD* (FlyBase ID FBgn0038134). *Cyp310a1* restricts Wingless expression to the dorso-ventral boundary (*Mohit et al., 2006*), whereas *wntD* is activated by dorsal and inhibited in the ventral cells by snail. *wntD* is also known to be regulated by dorsal/twist/snail and to inhibit dorsal nuclear localization and function (*Ganguly, Jiang & Ip, 2005*). Additionally, we detected dorsal and snail binding sites in the

promoter of *Neu2* (FlyBase ID FBgn0037085), which has been described as a dorsal target in the neuroectoderm (*Stathopoulos et al., 2002*). Other dorso-ventral target genes involved in morphogenesis and reported to contain high dorsal occupancy were also found. For example, dorsal sites were discovered in the promoter of *scra* (FlyBase ID FBgn0261385), but snail sites were instead detected in the promoters of *lbk* (FlyBase ID FBgn0034083) and *Zasp52* (FlyBase ID FBgn0265991) (*Koenecke et al., 2016*). These results demonstrate that at least for both transcription factors (dorsal and snail), our models were able to detect sequence motifs that lay inside binding occupancies.

To further evaluate the biological significance of our informative features, we collected genes whose promoters contained common structural features, and checked for shared molecular functions or biological processes according to the FlyBase repository (*Marygold et al., 2013*). Figure 7A shows the promoter regions of three genes expressed in adult male eclosion +5 days, and involved in motor neuron axon guidance, signal peptide and developmental protein. The promoters of genes Notch (FlyBase ID FBGN0004647) and Wnt oncogene analog 4 (FlyBase ID FBGN0010453) shared the positioning of motif DM-6 from 500 to 600 bp relative to the TSS on minus strand, and the binding order of motifs DM-6 (on minus strand) and DM-1 (on plus strand) upstream of the TSS. Moreover, the promoters of Notch (FlyBase ID FBGN0004647) and Plexin B (FlyBase ID FBGN0025740) contained the motif DM-1 on plus strand and downstream of the TSS. Figure 7B depicts the promoters of three genes expressed in embryo 16–18 h, and involved in immunoglobulin-like domain/fold, immunoglobulin subtype 2 and immunoglobulin subtype/domain. These regulatory regions comprise the presence of motifs DM-9 and DM-10 on minus strand and upstream of the TSS. Figure 7C illustrates the regions of three genes expressed in embryo 22–24 h and involved in neurogenesis. Their promoters contain the presence of motif DM-8 on minus strand and upstream of the TSS, and the binding order of motifs DM-8 (on minus strand), DM-8 (on minus strand) and DM-11 (on plus strand) upstream of the TSS. It is worth noting that the gene Notch is known to be involved in the Notch signaling pathway as well as in promoting neural differentiation. We then checked how many genes involved in the Notch signaling our model would be able to find in a genome-wide search. Surprisingly, we could detect not only well-known genes such as Notch, mind bomb 1 and extra macrochaetae, but also others recently reported in an interaction network for Notch signaling (*Mummery-Widmer et al., 2009*) (Fig. S1). We further looked for enrichment of other pathways supported by gene ontology terms. As a result, terms such as lateral inhibition, developmental protein, imaginal disc-derived wing morphogenesis, R8 cell fate commitment, peripheral nervous system development, phosphoprotein and protein binding were statistically significant (Table S8). A collection of predicted motifs in the promoters of high-scoring genes per developmental stage can be found in Data S2.

## CONCLUSIONS

This study has introduced two new structural features to a previously developed model—the presence of motifs and the relative distance of motif pairs to the TSS—for modeling the promoter regions of co-expressed genes. From these two features, the presence of

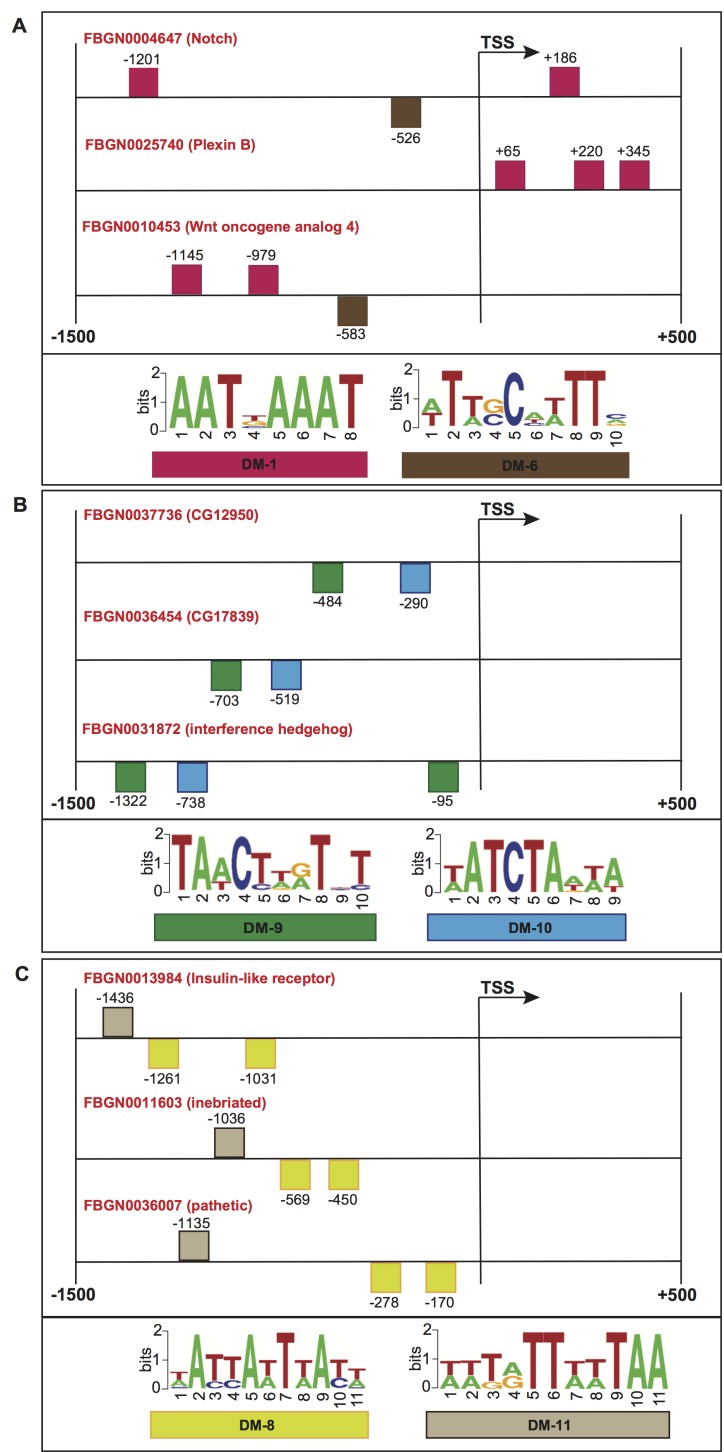

**Figure 7** Schematic representation of the promoter regions of three genes involved in (A) motor neuron axon guidance, signal peptide and developmental protein, (B) immunoglobulin-like domain/-fold, immunoglobulin subtype 2 and immunoglobulin subtype/domain, and (C) neurogenesis. Squares above/below the horizontal line indicate the DNA strand where the motif is located. For more detailed descriptions of these promoter regions, please refer to Data S2.

motifs without positional restriction, along with characteristics such as motif orientation, positioning of motifs relative to the TSS and pairwise distance of motifs appear to provide an accurate description of promoter regions. When the distance of two motifs relative to the TSS was included, there was not a significant improvement in performance. Features such as orientation, positioning of individual motifs relative to the TSS and pairwise distance of motifs were frequently included in the scoring scheme. The existence of common structural patterns in promoters of co-expressed genes was also confirmed by independent RNA-seq samples, resulting in 19 (79.2%) statistically significant models. Future studies should focus on simpler features in order to avoid complex promoter structures, which might not properly describe regulatory regions.

## ACKNOWLEDGEMENTS

The authors thank the members of Nakai Laboratory for their helpful comments and suggestions. Computation resources were provided by the Supercomputer System, Human Genome Center, The Institute of Medical Science, The University of Tokyo.

### Funding

This work was partly supported by Grants-in-Aid for Scientific Research from JSPS (16H04724 & 17K00397). Yosvany López was supported by the MEXT scholarship. The funders had no role in study design, data collection and analysis, decision to publish, or preparation of the manuscript.

### Grant Disclosures

The following grant information was disclosed by the authors:
JSPS: 16H04724, 17K00397.
MEXT scholarship.

### Competing Interests

Kenta Nakai is an Academic Editor for PeerJ.

### Author Contributions

- Yosvany López conceived and designed the experiments, performed the experiments, analyzed the data, wrote the paper, prepared figures and/or tables.
- Alexis Vandenbon reviewed drafts of the paper, gave a number of comments for improving the manuscript.
- Akinao Nose reviewed drafts of the paper, gave comments as an expert in fly development.
- Kenta Nakai conceived and designed the experiments, reviewed drafts of the paper, gave a number of comments for improving the manuscript.

### Supplemental Information

Supplemental information for this article can be found online at http://dx.doi.org/10.7717/peerj.3389#supplemental-information.

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
