# Peer review of "Modeling the cis-regulatory modules of genes expressed in developmental stages of Drosophila melanogaster"

_PeerJ, doi:10.7717/peerj.3389_

## Round 0.1 · original submission · Major Revisions

Please carefully consider the points raised by the reviewers. In my opinion they will be extremely important to improve the manuscript.

·

Basic reporting

The submission adheres to PeerJ policy with minor changes required in English.

Experimental design

The approach used by the authors is interesting and novel for the field. The biological relevance of the findings could be better discussed along the manuscript.

Validity of the findings

The data is controlled and statiscally sound, the appropriate statistical tests were performed. The unique exception regards the analysis of the regulatory motifs along the Drosopihila phylogenetic tree, which was quite scarce and not further explored.

Additional comments

Yosvany López, Alexis Vandenbon, Akinao Nose , Kenta Nakai

The manuscript “Modeling the cis-regulatory modules of genes expressed in
developmental stages of Drosophila melanogaster” by Lopez et al. performs a computational analysis using structural properties of promoter molecules. Six types of structural features have been analyzed by the authors. RNA-sequencing data of 22 developmental stages was used to create and validate the statistical models. Although the manuscript addresses the important question of gene regulatory prediction based only on sequence features and the approach applied is quite interesting, data presentation could be improved and the possible biological insights better discussed. Major and minor points are highlighted below.


Major points:

1) The whole validation manuscript analysis is based on RNA-seq data from modENCODE RNA obtained by different developmental stages, but of complete embryos, larval, pupal or adult tissues. It is widely known that during these developmental stages of Drosophila different classes of cis-regulatory elements (CREs) exist in every tissue, germ layer or organ (see for example Stark, A; Levine M.; or E. Furlong’s CRE papers). Thus, the assumption “we defined sets of genes with preferential expression in several D. melanogaster developmental stages. It is reasonable to assume that the promoters of such genes share similar structural patterns” is not valid for the dataset used (RNA-seq of complete individuals) since every RNA-seq was done with complete animals, thus with several tissues and cell types. Thus every RNA-seq stage will contain a mixture of different CREs, which can then be separated into distinct groups based on the occurrence of different motifs which are essentially regulated by different TFs. Thus, I would rephrase the whole sentence or even perform new analysis using known CREs for data validation (see below).
2) I am particularly concerned with the direct connection between levels of expression (FPKM) at every developmental stage and that the overrepresented motifs detected by the authors during the current study are overlapping with CREs. There has been a large number of characterized and deposited CREs e.g. in REDFLY. Thus, it would be extremely interesting to investigate if the overrepresented motifs identified by the current study lies inside annotated CREs or databases.
3) The addition of the phylogenetic tree with eleven Drosophila sibling species was not discussed or explored in detail. Only one regulatory region is discussed in the main text and two in the supplemental data. Some fly species do not show the predicted motifs, others present more copies of the motif but no biological explanation or evolutionary discussion or hypothesis is provided. Thus, I suggest that either the evolutionary analysis is removed or properly explored at the genomics level.

Minor points:

1) Abstract: I would remove the last sentence about ChIP-sequencing data, since the models presented here are still under development and it is not completely clear if they would apply to ChIP-Seq.
2) Page 3 row 56-57: “In addition, available ChIP-seq…..” (References missing).
3) Page 4 row 62. “The third instar larvae changes to pupae” - “changes” is not the appropriate word.
4) Page 4 rows 64-65. “The former…..internal organs of the adults” - Reference missing.
5) Page 4 rows 68-69. “D. melanogaster embryo during development has been integrated at the genome-wide scale” Reference missing
6) Page 5 rows 86-89. “Although these models…..suggesting” – Sentence is long and unclear.
7) Page 7 – row 138. Why sets with at least 55 genes were retained for further analysis?
8) Page 14. The authors should be more precise at the definition motif identification. For instance - Row 300 “which resembles the binding motif of regulator sna” AND row 303 – “in stage embryo 12-14h that shows similarity”. It would be important to state how this resemblance and similarity were calculated or estimated.
9) Row 312. “To check whether…..support the fact…” – This sentence is not clear.
10) Row 315 – “most of the genes encoding” – numbers ? It would be important to be specific.
11) Row 329 – “the model of embryo 2-4h, on the other hand, showed the worst performance”. At this stage, during fly embryogenesis the different regulatory networks responsible for AP and DV patterning and are established in an intense zygotic transcription mechanism involving different types of transcription factors and CREs. Couldn’t it explain this lack of performance? A differential number of CREs being activated at the same time ? Is the number of expressed genes different at that embryonic stage when compared to other stages ?
12) Page 17 - Although the findings of the specific motifs (DMs) along different stages is interesting, it was not clear to me their how similar their sequences are to published motifs. I could not find a table with them and their correlation with known motifs.
13) Page 19. The authors set the threshold of expression in FPKM >1. Did the authors analyzed how an increase into FPKM >2 or >3 would affect their results?
14) Page 20. Row 413. “ having them” ?
15) Page 20-21. The overrepresentation of Notch pathway members in the 10-12h analysis is quite interesting. Did the authors look for enrichment of other pathways or processes in any other dataset, e.g. GO ?
16) As said above the presence and absence of the two motifs in different positions and species - Figure 6 (phylogenetic tree) do not provide any biological insight.

Reviewer 2 ·

Basic reporting

Lopez et al. predict promoter architectures specific for preferentially expressed genes at 22 different stages of Drosophila development. The Authors eliminated promoters that are more than 60% similar. Then, preferentially expressed gene sets were subdivided into (1) motif-prediction, (2) feature-computation, and (3) model-construction subsets. Features included order, orientation, positioning of motifs relative to the TSS, pairwise distance of motifs, presence of motifs without position restriction and distance of motif pairs relative to the TSS. A genetic algorithm constructed the models to maximize fitness using crossing overs and mutations. At the end, a five-fold cross-validation was performed. This information should be present in the Abstract or at least, in the general section of the Methods.

Experimental design

P. 6. Lines 121-123: “The resulting datasets were separately mapped to the D. melanogaster genome (r6.02) with Bowtie 2 (-no-unal) (Langmead & Salzberg 2012) and TopHat2 (default parameters) (Kim et al. 2013).” tophat2 automatically calls bowtie2, is this what the Authors mean? If not, why is it necessary to call bowtie2 manually?
P. 6. The first few paragraphs of “Materials and Methods” about motif discovery should go to the detailed description of motif discovery.
P. 6. The identification of differentially expressed genes is not acceptable: replicates should not be merged due to losing all information on reproducibility (between replicates), inability to use proper statistical tests, etc. It is very well known that transcript levels do not follow the normal distribution but can modeled reasonably well by the negative binomial distribution. Hence Z-scores cannot be used. This has been implemented into generally accepted tools for analyzing differential expression such as DESeq2 and edgeR, downloadable from bioconductor.org.
P. 7, lines 141-145: What is the meaning of word length in transcript levels?
P. 9., line 181: Were the p-values corrected for multiple tests by using, for example, False Discovery Rate? This is not obvious for me from the supplementary file compute_p_values.py.
P. 13. The Authors collect a large but unknown numbers of Fscore-s and calculate the same number of p-values. These p-values must be corrected for multiple tests. For example, if the p-value threshold is set to 0.01 and 10,000 test are performed, purely by chance, 0.01 * 10,000 = 100 false positive results are expected. Performing multiple test correction is an absolute condition for accepting the manuscript.
P. 16, Table 1: Please perform multiple test corrections! Also, compute_p_values.py should be modified to provide higher accuracy than 10-7.
In addition to the genetic algorithm, the Authors, at their discretion, may consider performing permutation tests by comparing predicted positives and negatives to a set of permutations, where individual promoters are randomly assigned to positive and negative sets.

Validity of the findings

The above methods are elegant and interesting. My enthusiasm is muted however, as the Authors do not perform reality check by comparing predicted promoter architectures and models to ChIP-seq observations. Even though ChIP-seq data contain numerous false positive and false negative observations, they provide the most realistic assessment of transcription factor binding at the current state of the art. All models should be compared to real-life evidence if available, but this model in particular, for several specific reasons:
1. Binding sites of transcription factors are short and degenerate, hence many of the motifs do not bind regulators at a given time and in a given cell.
2. Many transcription factors bind to very low-scoring motifs or non-motifs.
3. Histone modifications are not considered.
4. Regulatory interactions are only implicitly modeled.
5. Distant enhancers are not considered in this model. P. 5, lines 89-90: The statement that “complex structural patterns might not be so important” contradicts to major chromosomal structure observations obtained by chromosomal conformation capture methods and the confirmed action of distal enhancers, neither of them mentioned here.
6. No selection is involved in the model in any direct way.
In contrast, the Authors claim that “To some extend,” [extent] “our method could be applicable to search for genes expressed in D. melanogaster developmental stages without the necessity of ChIP-sequencing data.” However, to justify this claim, the Authors should compare their results to the already available vast amount of ChIP-seq data in D. melanogaster and in other organisms. This would be a real-life validation of their results, absolutely critical to the message of the manuscript.

Additional comments

Promoter architectures and models should be communicated in machine-readable formats, not only in figures.
P. 6. The first few paragraphs of “Materials and Methods” about motif discovery should go to the detailed description of motif discovery.
P. 10., line 233. Please fix the typo in Eq. 2, I it not “F – score” but Fscore (score is subscript).
P. 12, line 246: please include the Kullback-Leibler measure in the formula.

In addition to the genetic algorithm, the Authors, at their discretion, may consider performing permutation tests by comparing predicted positives and negatives to a set of permutations, where individual promoters are randomly assigned to positive and negative sets.

Supplementary Figures S1 and S2 should go to the main text.
There are some minor language issues/typos:
P. 5, lines 104-105: “It first predicts de novo motifs and selects those overrepresented ones” – should be “the overrepresented ones”.
Please insert a comma after etc.
No first names in citations.
“Consortium Tm” is not an author. When using Endnote, please double-quote consortium.
FASTQC has already been published.

---

## Round 0.2 · Major Revisions

It is very important to address the the criticisms from Reviewer 2, in particular those regarding: 1) the use of a spliced aligner; 2) Differential expression analysis; 3) a solid rationale for merging biological replicates, which seems rather counterintuitive at the best.

·

Basic reporting

In this new version of the manuscript all my concerns have been properly addressed.

Experimental design

All suggestions required in the first round of review have been addressed by the authors.

Validity of the findings

After rewriting and reanalysis performed the manuscript has been largely improved and data presentation became quite clear.

Additional comments

In this new version of the manuscripts, the authors have largely developed the suggestions of this reviewer. Thus, I am happy with this new version of the manuscript.

Reviewer 2 ·

Basic reporting

Minor issues
• P. 3, lines 60-61 “The third instar larva molts into a pupa during which the larva metamorphoses into the adult fly.” When the larva transforms into a pupa, it becomes a pupa. The sentence should read “The third instar larva molts into a pupa, which morphs into an adult fly”.
• P. 7, line 143: “modENCODE Consortium.

Experimental design

Rather statistical and computational approaches

• The RNA-seq analysis remains way below the standards of the profession. This way the manuscript cannot be accepted. Such a publication is expected to be based on solid computational/statistical results and should communicate acceptable practices to younger generations of scientists.

o Rebuttal letter, page 6, par. 2:” We merged replicates following a similar procedure (Li et al., Genome Research 2014, 24: 1086-1101), which was developed to detect expressed genes in D. melanogaster developmental stages. We totally agree with the reviewer on this issue. However, it is in general difficult to define "condition-specific expression" so that in this case we opted for a simple approach.”
Just because some method was published it does not mean that it is acceptable. Please refrain from merging the biological replicates otherwise your statistical conclusions will be unacceptable.

o My original comment: “The resulting datasets were separately mapped to the D. melanogaster genome (r6.02) with Bowtie 2 (-no-unal) (Langmead & Salzberg 2012) and TopHat2 (default parameters) (Kim et al. 2013).” tophat2 automatically calls bowtie2, is this what the Authors mean? If not, why is it necessary to call bowtie2 manually?
The Authors reply: “We apologize to the reviewer for the confusion and thank him/her for pointing out this issue. The datasets were mapped with Bowtie 2 and TopHat2 was not used whatsoever. Therefore, we removed any reference to TopHat2 in the revised manuscript.”

Using tophat2 (or STAR or other spliced alignment tool) is necessary. The reason is that bowtie and other unspliced alignment tools cannot map those sequencing reads that join two exons over an intron. Taking into account the number of exons in Drosophila, using bowtie would leave a large number of biexonic sequencing reads unmapped. Please do use a spliced alignment tool such as tophat2! Then, you do not need to call bowtie2 manually.

o My original comment: “It is very well known that transcript levels do not follow the normal distribution but can modeled reasonably well by the negative binomial distribution. Hence Z-scores cannot be used. This has been implemented into generally accepted tools for analyzing differential expression such as DESeq2 and edgeR, downloadable from bioconductor.org.”

o My new comment. Strangely, no reply was provided! It is very well known that even logarithm-transformed transcript levels follow a very long-tailed distribution, which cannot and should not be simplified as normal. It is well known that if a long-tailed distribution is treated as if it were normal, you will get more “significant” differences, here, differentially expressed genes, most of them being false positives. Such approaches have been harshly criticized by Terry Speed, Gordon K. Smyth, and several others – back in 2003! Please perform a publishable differential expression analysis by DESeq2, edgeR, or any well-accepted method that does not rely on the normal distribution (as Z-scores do).

Validity of the findings

• Performance measurements, validations. My original comments:
QUOTE
The above methods are elegant and interesting. My enthusiasm is muted however, as the Authors do not perform reality check by comparing predicted promoter architectures and models to ChIP-seq observations. Even though ChIP-seq data contain numerous false positive and false negative observations, they provide the most realistic assessment of transcription factor binding at the current state of the art. All models should be compared to real-life evidence if available, but this model in particular, for several specific reasons:
1. Binding sites of transcription factors are short and degenerate, hence many of the motifs do not bind regulators at a given time and in a given cell.
2. Many transcription factors bind to very low-scoring motifs or non-motifs.”
END QUOTE
The Authors’s reply: “We totally agree with the reviewer’s comment and tried to perform real-life validation with ChIP-seq data. However, although the modENCODE repository contains a lot of datasets for D. melanogaster, only a few of our detected transcription factors were available. Instead, we downloaded experimentally validated cis-regulatory elements (CREs) from the REDfly database. These CREs consisted of Reporter Constructs and Transcription Factor Binding Sites (TFBS), tested by reporter gene assays or discovered by DNaseI footprinting and electrophoretic mobility shift assays. For each developmental stage we checked whether our predicted motifs related to high-scoring genes were located inside such CREs. This analysis was added to the sections “Materials and Methods” (pp. 13-14 lines 283-288) and “Results and Discussion” (pp. 20-21 lines 431-435). Additionally, the predicted motifs overlapping experimentally validated CREs are provided in the Supplemental Data S1.”

My new reply: modENCODE and other labs have published ChIP-seq data that provide an acceptable (not excellent) sample for validating the models. Even if the Authors use experimentally validated CREs, a real QUANTITATIVE validation would be necessary. Their new discussion of this validation:

“This analysis resulted in many of our predicted sites lying within experimentally validated elements (Data S1).”

This is neither quantitative nor acceptable. Data S1 consists of 22 tab-delimited text files with the coordinates of the predictions and the experimental results. No general statistics and no single overlap between predicted and experimental observations are shown. Performing a quantitative validation and interpreting it verbally are absolutely critical tasks. This validation is expected to show the utility and the veracity of the results obtained in this manuscript.

---

## Round 0.3 · accepted · Accept

The criticisms of the reviewers were largely addressed and the manuscript can be now accepted for publication.